# Predicting and Treating Pulmonary Fibrosis with Proteomic Biomarker Investigations

**DOI:** 10.3390/biomedicines13112656

**Published:** 2025-10-29

**Authors:** Giulia Raineri, Anna Valeria Samarelli, Roberto Tonelli, Valentina Masciale, Beatrice Aramini, Tiziana Petrachi, Giulia Bruzzi, Filippo Gozzi, Ester Trasforini, Angela Esposito, Filippo Azzali, Massimo Dominici, Albino Eccher, Stefania Cerri, Enrico Clini

**Affiliations:** 1Experimental Pneumology Laboratory, Department of Medical and Surgical Sciences, University of Modena and Reggio Emilia, 41125 Modena, Italy; giuliaraineri@unimore.it (G.R.); roberto.tonelli@unimore.it (R.T.); giuliabruzzi@unimore.it (G.B.); filippo.gozzi@unimore.it (F.G.); estersyncro@icloud.com (E.T.); angy_160402@libero.it (A.E.); albino.eccher@unimore.it (A.E.); stefania.cerri@unimore.it (S.C.); enrico.clini@unimore.it (E.C.); 2Respiratory Diseases Unit, Department of Medical and Surgical Sciences, University Hospital of Modena, University of Modena and Reggio Emilia (UNIMORE), 41125 Modena, Italy; 3PhD Program in Molecular and Regenerative Medicine, University of Modena and Reggio Emilia (UNIMORE), 41125 Modena, Italy; 4Laboratory of Cellular Therapy, Department of Medical and Surgical Sciences, University of Modena and Reggio Emilia (UNIMORE), 41125 Modena, Italy; valentina.masciale@unimore.it (V.M.); massimo.dominici@unimore.it (M.D.); 5Division of Thoracic Surgery, Department of Medical and Surgical Sciences, DIMEC of the Alma Mater Studiorum, University of Bologna, G.B. Morgagni-L. Pierantoni Hospital, 47121 Forlì, Italy; beatrice.aramini2@unibo.it; 6Tecnopolo Mario Veronesi, Via 29 Maggio 6, 41037 Mirandola, Italy; tiziana.petrachi@tpm.bio; 7Pathology Unit, University Hospital of Modena, 41125 Modena, Italy; azzali.filippo@aou.mo.it; 8Program in Cellular Therapy and Immuno-Oncology, Division of Medical Oncology, Residency School of Medical Oncology, 41125 Modena, Italy; 9Center for Rare Lung Diseases, University Hospital of Modena, 41125 Modena, Italy

**Keywords:** mass spectrometry, IPF, biomarkers, proteomic analysis, machine learning

## Abstract

Idiopathic pulmonary fibrosis (IPF) is a chronic, rare, and fatal disease that is the consequence of aberrant remodeling and defective repair mechanisms within the lung, culminating in the loss of alveolar integrity. Although significant progress has been made in understanding the pathogenesis, it would be crucial to identify biomarkers for diagnosis, prognosis, and prediction of therapy response to improve the management of this challenging and debilitating disease. Omics technologies have profoundly advanced the understanding of disease mechanisms, presenting considerable potential for the identification of clinically relevant biomarkers. To date, specific molecular pathways have been implicated in the onset and progression of idiopathic pulmonary fibrosis, including abnormal wounding, fibroblast proliferation, inflammation, deposition of the extracellular matrix, oxidative stress, endoplasmic reticulum stress, and the coagulation system. This review highlights the role of proteomics in identifying key biomarkers for IPF, focusing on their clinical relevance, including diagnosis, prognosis, disease progression, and the identification of new therapeutic options, in light of the most recent technological advancements in mass spectrometry.

## 1. Introduction

Idiopathic pulmonary fibrosis (IPF) represents the most common form of idiopathic interstitial pneumonia (IIP) and is characterized by the progressive distortion of the lung parenchyma structure, aberrant extracellular matrix (ECM) deposition, and irreversible scarring. It is a chronic and rare disease occurring in adults over 60 years old, with a median survival of 3 years after diagnosis, which is diagnosed according to multidisciplinary clinicopathological criteria, typically displaying the so-called usual interstitial pneumonia (UIP) pattern [1].

Although the cause of IPF onset is unknown by definition, several risk factors have been identified that may increase the likelihood of developing this disease [2,3], such as environmental exposure, radiation therapy, occupational exposure (e.g., asbestosis), cigarette smoking, and exposure to allergens.

The prognosis of IPF may be unpredictable, since patients can be characterized by progressive respiratory failure, often precipitated by acute events, namely, acute exacerbations (AE) [4]. Despite the introduction of drugs, such as Pirfenidone and Nintedanib, which can slow respiratory functional decline, IPF still has poor survival rates (20–40% after 5 years from diagnosis).

A crucial clinical need in IPF is the identification of reliable biomarkers for early detection, disease staging, prognosis, and therapeutic monitoring [5]. In this context, mass spectrometry (MS)-based proteomics has emerged as a particularly promising tool to elucidate protein-level alterations involved in fibrotic mechanisms. Unlike genomics and transcriptomics, which infer cellular activity from upstream nucleic acid levels, proteomics enables the comprehensive profiling of the proteome, including post-translational modifications (PTMs), protein degradation, and dynamic expression changes in response to environmental and pathological stimuli. These features are especially relevant in IPF, where aberrant protein signaling, ECM remodeling, and acute inflammatory events are hallmarks of disease progression and acute exacerbations. Moreover, MS-based proteomics is increasingly being applied to human biofluids, such as bronchoalveolar lavage (BAL) fluid, serum, and plasma, offering a non-invasive approach to biomarker discovery and patient stratification [6,7].

This review aims to provide an updated overview of MS-based proteomics workflows and techniques that may be used to understand the mechanisms underlying the onset, progression, and AE of pulmonary fibrosis, supporting personalized diagnostics and treatment. We highlight recent advances in single-cell and spatial proteomics, biofluid profiling, and integrative multi-omics strategies, emphasizing their potential to support the transition of proteomics from basic research to clinical diagnostics and personalized treatment in pulmonary fibrosis.

## 2. Diagnostic, Prognostic, and Therapeutic Challenges of IPF

IPF is diagnosed by clinicopathological criteria, including the radiographic and/or histopathological hallmark pattern of UIP [8]. Histologically, the UIP pattern is characterized by a heterogeneous phenotype with the presence of fibroblastic foci (FF) interspersed with regions of relatively preserved parenchyma, as well as areas of inflammation and honeycombing. High-resolution computed tomography (HRCT) of IPF patients shows a peripheral, subpleural, and basal distribution pattern characterized by reticular opacities, honeycombing, and architectural distortion [9]. The definition of IPF requires the exclusion of other forms of interstitial pneumonia, including other idiopathic interstitial pneumonias and ILD associated with environmental exposures, medications, or systemic diseases. Since the histological pattern is not exclusively associated with IPF but also with other ILDs, establishing a definitive diagnosis remains challenging.

The pathogenesis of IPF is complex and far from being fully understood. Much evidence suggests that IPF is a consequence of multiple interacting genetic and environmental risk factors, such as cigarette smoking, metal and silica dust, or microbial agents, leading to repetitive local micro-injuries to the alveolar epithelium and vascular endothelium, which cause aberrant epithelial–fibroblast functional activity, excessive ECM deposition, and aberrant tissue repair processes. These pathological mechanisms involve the activation, proliferation, and differentiation of fibroblasts into myofibroblasts and the secretion of profibrotic, coagulant, and inflammatory cytokines, ultimately resulting in lung architectural distortion and a progressive decline in respiratory function [10,11]. In addition, some genetic variants have also been associated with IPF onset and progression, such as variants in the *mucin 5B (MUC5B)* gene, which plays a role in bronchoalveolar epithelial function, and in the *TERT* and *TERC* genes, which are involved in telomere-length maintenance [7]. Nevertheless, IPF remains a fatal disease, and the identification of different and new biomarkers indicative of progression represents an unmet clinical need that has to be addressed.

The natural course of IPF is variable, and survival can be influenced by multiple clinical factors, including the frequency and severity of AE, comorbidities (e.g., lung cancer, emphysema, pulmonary hypertension), and the rate of functional decline [4].

Therapeutic options remain a clinical challenge. Two antifibrotic agents, Pirfenidone and Nintedanib, have demonstrated efficacy in slowing FVC decline and disease progression, although neither has been shown to significantly reduce mortality.

Pirfenidone has antifibrotic, anti-inflammatory, and antioxidant effects, primarily through the inhibition of transforming growth factor-beta (TGF-β) signaling. Common adverse effects include photosensitivity, gastrointestinal disturbances, and nausea. Nintedanib, a non-selective tyrosine kinase inhibitor targeting PDGF, VEGF, and FGF receptors, is also associated with gastrointestinal side effects, particularly diarrhea, nausea, and vomiting [12]. As a therapeutic strategy, the two antifibrotic drugs, Pirfenidone and Nintedanib, can slow the decline in respiratory function of IPF patients, and, according to both real-world data and randomized controlled trials, mortality rates and survival times in IPF are highly heterogeneous [13]. Emerging therapies offer potential hope. Recently, BI 1015550 (Nerandomilast), an oral selective phosphodiesterase 4B (PDE4B) inhibitor with antifibrotic and immunomodulatory properties, demonstrated efficacy in a Phase 2 trial by stabilizing lung function over a 12-week period. Furthermore, a Phase 3 randomized, double-blind, placebo-controlled trial reported that Nerandomilast significantly reduced the decline in FVC over 52 weeks compared to a placebo [14].

Supportive measures, including long-term oxygen therapy in cases of hypoxemia and pulmonary rehabilitation in functionally disabled individuals, are essential components of comprehensive care.

## 3. Protein Analysis Through Different Types of Mass Spectrometry Relevant for IPF Research

A robust proteomic study must be based on the accurate definition of study cohorts and the meticulous design of the experimental workflow.

The selection of an appropriate analytical strategy depends on multiple factors, including the biochemical properties of the target protein, the depth of proteome coverage required, and the need for absolute versus relative quantification. Mass spectrometry (MS)-based proteomics has emerged as a pivotal tool in the search for reliable biomarkers in IPF. Among the available platforms, liquid chromatography–mass spectrometry (LC-MS) has proven particularly effective due to its sensitivity, quantitative capabilities, and compatibility with complex biological samples, such as bronchoalveolar lavage fluid (BALF), plasma, and lung tissue. LC-MS is particularly suitable for analyzing a wide range of biomolecules, including peptides and intact proteins [15], and has been extensively used to identify protein biomarkers for diagnostic, prognostic, and therapeutic purposes in IPF. Protein identification can be achieved by a bottom-up approach (Figure 1A), where peptides derived from enzymatic digestion are analyzed, or by a top-down approach (Figure 1B), where intact proteins are examined directly. This latter strategy provides more detailed sequence coverage and information about post-translational modifications (PTMs) [16,17]. Top-down proteomics typically involves separating proteins by one- or two-dimensional liquid chromatography, followed by identification using high-resolution mass spectrometry, facilitating the detection of proteoforms resulting from PTMs, genetic mutations, RNA splicing, and histone modifications and other multisite heterogeneities.

The main limitation of the top-down proteomic approach for broader application is related to the lower solubility of intact proteins compared to their peptide counterparts [18]. In contrast, bottom-up proteomic approaches are widely applied because peptides are easier to separate and analyze by LC coupled to tandem mass spectrometry (MS/MS) [19]. The combined approach enables a detailed analysis of both intact proteins and peptide fragments, enhancing the identification of PTMs and isoforms. In addition, gas chromatography–mass spectrometry (GC-MS) has been used to analyze volatile organic compound (VOC) profiles in IPF patients, patients with different ILDs, and healthy controls. In their study, Plantier et al. identified significantly higher concentrations of acetone and other VOCs in IPF patients, suggesting metabolic alterations associated with oxidative stress and inflammation [20,21].

Furthermore, quantitative proteomics methods, such as isobaric tags for relative and absolute quantitation (iTRAQ), enable the simultaneous comparison of multiple biological samples by labeling peptides with distinct isotopic tags, thereby allowing relative abundance measurements across experimental conditions [22]. In addition to MS methods, gel electrophoresis techniques, such as two-dimensional gel electrophoresis (2D), remain valuable for assessing differential protein expression patterns [23].

Finally, high-resolution mass spectrometry (HRMS) can employ electrospray ionization (ESI) or matrix-assisted laser desorption ionization (MALDI). ESI is particularly suitable for analyzing proteins in solution due to its ability to generate high-quality ionized peptides, while MALDI allows for rapid analysis of complex tissue samples, making it a valuable tool in pulmonary research, as it enables direct spatial localization of ECM alterations [24].

Thus, the combined use of advanced proteomic technologies, high-resolution mass spectrometry, quantitative proteomics, and bioinformatics analysis has enabled the characterization of the complex protein networks involved in rare lung diseases, like IPF, across different biological samples. In the next sections, we will summarize the biomarkers of IPF onset and progression identified through mass spectrometry in different biological samples.

## 4. Mass Spectrometry-Based Serum and Plasma Biomarkers in Idiopathic Pulmonary Fibrosis

Since the diagnosis of IPF is often complex and invasive, the identification of non-invasive biomarkers could simplify the diagnostic process and improve access to care for patients [25]. The detection of circulating biomarkers in the serum or plasma of patients affected by IPF offers non-invasive tools for diagnosis, outcome prediction, and treatment monitoring, aiming to prevent significant clinical and functional decline [26]. One of the most robust and reproducible circulating biomarkers is matrix metalloproteinase-7 (MMP-7) [27]. MMP-7 belongs to the family of zinc-dependent endopeptidases and plays a pivotal role in degrading ECM components, such as collagens, fibronectin, and laminin [28]. The critical role of this protein in fibrosis has been demonstrated in MMP-7 knockout mice, which exhibited strong resistance to developing a fibrotic phenotype following intratracheal administration of bleomycin [29]. The proteolytic activity of MMP-7 in lung fibrosis, together with its elevated plasma levels in IPF patients, suggests its utility as a biomarker for discriminating between healthy and diseased individuals. Moreover, it may aid in the differential diagnosis between IPF and other lung diseases, such as chronic obstructive pulmonary disease (COPD), sarcoidosis, and hypersensitivity pneumonitis [30].

In addition to MMP-7, other matrix remodeling proteins (such as MMP-1 and MMP-10) have also been investigated as potential disease biomarkers [31]. Collagen production and abnormal ECM deposition—hallmark features of fibrosis—are closely associated with CC-chemokine 18 (CCL18), a chemokine that exerts chemotactic and immunoregulatory functions. Elevated serum levels of CCL18 have been consistently reported in patients with pulmonary fibrosis, with studies demonstrating a correlation between serum concentrations and the extent of fibrotic activity. CCL18 is also associated with organ failure and, as a predictor of survival and disease progression, is considered a promising prognostic marker in IPF [32,33]. Furthermore, high serum levels of periostin—a protein involved in the ECM remodeling and linked to pulmonary function decline—have been observed in IPF patients [34]. Once secreted by fibroblast, epithelial, and endothelial cells, periostin interacts with integrin receptors following stimulation by IL-4, IL-13, and TGF-β, thereby perpetuating fibrosis [35,36]. Several studies have also shown that patients with IPF exhibit elevated serum levels of surfactant proteins SP-A and SP-D compared to healthy controls [26]. These proteins, involved in maintaining tissue homeostasis and immune response, contribute to differential diagnosis and prognosis prediction [33]. Krebs von den Lungen-6 (KL-6) is a high-molecular-weight mucin-like glycoprotein produced by damaged and regenerating alveolar type II cells in the lungs [37]. KL-6 has been proposed as a diagnostic marker for interstitial lung diseases (ILDs) and as a potential predictor of response to antifibrotic therapy.

In recent years, advances in proteomics have led to the identification of novel blood-based biomarker panels for IPF (Figure 2). For example, the SOMAmer-based proteomic platform (SOMAscan) was employed to analyze plasma from 60 IPF patients over an 80-week follow-up period. The study identified a panel of six analytes associated with disease status. Specifically, disease progression was associated with sub-threshold levels of soluble vascular endothelial growth factor receptor 2, ficolin-2, legumain, and cathepsin, along with elevated levels of inducible T cell co-stimulator and trypsin-3 [38].

Additionally, the iTRAQ method, which enables simultaneous identification and quantification of reporter ion peak intensities using tandem mass spectrometry (MS/MS), revealed the presence of four analytes associated with IPF. In the studied cohort of patients, elevated levels of C-reactive protein (CRP) and fibrinogen-α, along with decreased levels of haptoglobin and kininogen-1, were observed in comparison with healthy individuals. These biomarkers, identified through proteomic studies, provide insights into IPF pathogenesis and hold potential for improving diagnosis, prognosis, and treatment monitoring. However, further validation is needed before they can be routinely used in clinical practice [39]. Moreover, Oldham et al. employed quantitative proteomics to analyze plasma samples from IPF patients and identified significant differences in protein expression between progressive and stable disease. Key biomarkers included increased levels of serum amyloid A1 (SAA1), haptoglobin (HP), and hemopexin (HPX), which were involved in inflammatory response, oxidative stress, and ECM remodeling—processes critical in IPF pathogenesis [40]. While circulating biomarkers offer potential for diagnosing and managing IPF, challenges such as a lack of specificity and the need for standardization persist. The use of biomarker panels may enhance diagnostic and prognostic accuracy by capturing the multifactorial nature of the disease. Longitudinal studies are essential to assess their predictive value, and their integration with advanced technologies could further optimize diagnostic precision and treatment monitoring. Finally, although several promising biomarkers—such as MMP-7, CCL18, KL-6, and periostin—have been identified in the studies mentioned above, many are limited by small sample sizes and lack external validation or replication in independent cohorts. Most studies are observational and cross-sectional, with few longitudinal or interventional designs, making it difficult to establish causality or predictive utility. Furthermore, confounding factors, such as comorbidities, smoking status, medication use, and disease heterogeneity, are often not adequately addressed, thereby limiting their clinical applicability.

## 5. Biomarkers of IPF Identified in BALF Fluid Through Proteomics

BALF has been increasingly used to identify disease-associated protein signatures in IPF, as it reflects the lung microenvironment, offering insights into local immune responses and tissue remodeling processes, while involving minimal invasiveness compared to surgical biopsy. Advances in proteomic technologies and MS have enabled the identification of numerous proteins in BALF that are differentially expressed in IPF patients compared to healthy individuals or those with other interstitial lung diseases (Figure 3). Elevated levels of mediators of pulmonary fibrosis—including osteopontin, MMP7, CXCL7, CCL18—as well as eosinophil and neutrophil-derived proteins and proteins associated with fibroblast foci, have been observed in BALF from IPF patients using gel-free quantitative proteomics (HDMSE) and targeted multiple reaction monitoring (MRM). Additionally, one study described, for the first time, the upregulation of the profibrotic cytokine CCL24 in the BALF of IPF patients [41]. Furthermore, a comparison of BALF protein profiles between IPF patients, never-smoker healthy controls, and smoker controls using comparative two-dimensional gel electrophoresis (2D-PAGE) analysis revealed that transcription factors NF-kB, PPARγ, and c-MYC act as functional hubs, indicating the principal pathways involved in IPF progression and pathogenesis [42]. In one study, 2D-PAGE combined with MALDI analysis uncovered the overexpression of proteins, such as S100A9, in the BALF of patients with IPF, as compared to patients with other fibrotic diseases, highlighting the value of proteomics in identifying biomarkers capable of distinguishing among different types of fibrotic interstitial pneumonia [43]. Another comparative analysis of BALF protein composition in patients with IPF, sarcoidosis, and pulmonary fibrosis associated with systemic sclerosis (SSc) using 2-DE showed that quantitative rather than qualitative differences characterized the protein profiles. Specifically, BALF from IPF patients displayed a higher abundance of low-molecular-weight proteins compared to those from patients with sarcoidosis and SSc. In addition, cytokine-focused proteomic studies reported elevated levels of IL-1, IL-6, IL-8, TNF-α, and IL-17A/F, alongside increased cellular counts of neutrophils, macrophages, and eosinophils, in BALF from IPF patients. Furthermore, macrophage migration inhibitory factor (MIF), along with p23 and Calgranulin B, two calcium-binding proteins, were also significantly increased in IPF [44]. Finally, label-free quantitative (LFQ) proteomic profiling of BALF, combined with unsupervised cluster analysis, identified protein expression patterns that correlate with survival outcomes in patients with idiopathic interstitial pneumonias (IIPs), reinforcing the prognostic value of BALF biomarkers [45].

In summary, a wide array of BALF-derived biomarkers—identified through high-throughput proteomics and MS techniques, including HDMSE, 2D-PAGE, MALDI-ToF/ToF, LC-MS/MS, and MRM—has elucidated key proteins involved in inflammation, ECM remodeling, cell signaling, and immune modulation in IPF. These findings underscore the potential of BALF proteomics for disease monitoring, prognostication, and development of personalized therapeutic strategies—pending validation in larger, multicenter studies. Thus, although BALF proteomics has uncovered numerous disease-associated proteins in IPF, most studies are limited by small sample sizes and single-center, exploratory designs. Replication in independent cohorts remains rare, and confounding factors, such as smoking status, co-existing ILDs, and treatment history, are often inadequately controlled, limiting the generalizability and clinical translation of these findings.

## 6. Biomarkers of IPF Identified in Lung Tissues Through Proteomics

Over the last few years, various attempts have been made to understand the molecular mechanisms underlying IPF. Among these, tissue-based proteomics has had a particularly significant impact, offering a comprehensive overview of the molecular pathways involved in the onset and progression of the disease. Given the rarity of the disease and the challenges associated with obtaining fresh samples, such as surgical biopsies or cryobiopsies, FFPE tissue specimens from pathology archives offer a reliable and accessible alternative for proteomic investigations and biomarker discovery [46]. However, formalin fixation and paraffin embedding can affect protein quality, thereby limiting the efficiency of proteomic analysis [47]. Despite these difficulties with FFPE samples, they have been successfully used for advanced proteomic analysis. In a study by Samarelli et al., FFPE lung tissue samples from patients with IPF were analyzed using LC-MS/MS and label-free quantitative proteomics, leading to the identification of differentially expressed proteins. These included proteins involved in the ECM signaling, focal adhesion, and transforming growth factor β (TGF-β) signaling pathways, all strongly associated with IPF pathogenesis when compared to nonfibrotic lung tissue used as controls. Moreover, five proteins were significantly overexpressed in the lungs of IPF patients with either an advanced disease stage (Stage II) or impaired pulmonary function (FVC <75%, Diffusion Lung Carbon Monoxide DLCO <55%) compared to controls. These proteins—lymphocyte cytosolic protein 1 (LCP1), peroxiredoxin-2 (PRDX2), transgelin 2 (TAGLN2), lumican (LUM), and mimecan (OGN)—may play key roles in fibrogenesis and represent potential biomarkers of disease progression [48]. Thus, the analysis of FFPE samples holds promise for identifying key proteins in IPF pathogenesis and progression.

Several recent studies have applied proteomic analysis to fresh or fresh-frozen lung tissue from IPF patients, identifying a more comprehensive proteomic profile than that obtained from FFPE samples. Among these, the application of iTRAQ labeling combined with LC-MS/MS to IPF fresh-frozen lung tissues revealed over 600 differentially expressed proteins, including well-studied ECM components—such as COL1A1, SCGB1A1, TAGLN, CTSB, and SERPINB3—as well as novel ECM-associated proteins like LGALS7, ASPN, and HSP90AA1 and HSP90AB1 [49].

One large-scale quantitative proteomic study detected more than 1500 differentially expressed proteins in IPF lungs compared to controls [50]. Interestingly, while transcriptomic signatures in IPF lungs were enriched in immune-mediated processes and inflammatory response pathways, protein expression profiles of IPF lung tissues were predominantly associated with ECM production, deposition, and remodeling. Two candidate proteins, namely, BTNL9, crucial in the process of antigen presentation, and PLLP, involved in epithelial cell development and differentiation, were downregulated at the mRNA and protein levels in IPF lung tissue and bleomycin-induced mice, indicating that they have a protective effect by inhibiting ECM production and promoting wound repair in alveolar epithelial cells [50]. A comprehensive summary of these biomarkers is shown in Figure 4. Given the histological heterogeneity in lung tissue regions in patients with ILDs, bulk tissue proteomic analysis—whether performed on fresh-frozen or FFPE samples—may not be sufficient to identify niche-specific biomarkers in IPF.

### New Insight into Biomarker Identification Through Spatial Proteomics

Spatial proteomics has recently emerged as a powerful approach for dissecting the molecular complexity of IPF by mapping protein expression across histologically distinct regions of lung tissue. In particular, spatially resolved proteomic investigations have contributed to the identification of niche-specific biomarkers in IPF, providing novel insights into disease progression and potential region-targeted therapies.

Laser capture microdissection coupled with mass spectrometry (LCM-MS) has been employed to isolate specific fibrotic niches—such as fibroblast foci (FF), fibrotic alveoli, and scarred regions—from IPF lungs. A study by Herrera et al. [51] used LCM-MS to analyze FF, adjacent mature scars, and adjacent alveoli in six fibrotic (UIP/IPF) and six nonfibrotic alveolar control specimens, identifying over 3000 proteins, with fibrotic alveoli characterized by a protein signature indicative of immune deregulation. Moreover, FF were positive for both transforming growth factor beta 1 TGFβ1 and TGFβ3, while the aberrant basaloid cell lining of FF was predominantly positive for TGFβ2. A recent integrative study combining spatial proteomics and single-cell RNA sequencing (scRNA-seq) revealed that FF are enriched in profibrotic proteins, such as latent TGF-β binding protein 1 (LTBP1) and fibronectin (FN1), confirming their active role in TGF-β signaling and ECM remodeling [52]. Furthermore, spatial proteomics of the basement membrane of distal bronchioles of IPF patients versus controls revealed the downregulation of cell junctional proteins, the upregulation of epithelial–mesenchymal transition (EMT) factors, and altered basement membrane matrix composition, resulting in epithelial desquamation in distal bronchioles [53]. Additionally, Griesser et al. applied quantitative proteomics to whole-lung homogenates from an AAV-DTR mouse model of acute epithelial lung injury, recapitulating IPF pathogenesis. By comparing bulk tissue data with spatially resolved epithelial region proteomes from the same animals, they observed injury-induced upregulation of pathways related to interferon responses, cell proliferation, DNA replication, and ECM deposition, along with the downregulation of epithelial markers (SP-A, SP-C, SCGB1A1), cilia assembly, lipid metabolism, and redox homeostasis. Their study, based on a mouse model, showed that the comparison between two proteomic methods revealed both extensive overlap and notable differences, highlighting that while bulk proteomics is broadly informative, region-specific analyses add valuable resolution [54].

Thus, advances in proteomic technologies have facilitated the application of mass spectrometry across a spectrum of tissue samples—from FFPE to fresh or fresh-frozen lung tissue—enhancing the understanding of biomarkers associated with IPF progression, derived from BALF and blood analyses. To assess the molecular contribution of different cell populations within the lung niche, proteomic analyses have been applied to cell lines and primary cells from patients with IPF.

## 7. Biomarkers of IPF Identified in Pulmonary Cell Lines and Primary Cells from IPF Patients

Cell-based proteomic analysis enables the analysis of protein expression dynamics, post-translational modifications, and secretion profiles at a functional resolution. Mass spectrometry-based studies on IPF-derived fibroblast cell lines, lung fibroblast cell lines, LL97A, and LL29 have revealed distinct biomarkers and molecular pathways implicated in fibrosis. In a label-free LC-MS/MS study comparing these IPF fibroblast cell lines with the normal lung fibroblast cell line (CCD19Lu), 80 proteins were found exclusively expressed in IPF cells, with an additional 19 upregulated in LL97A and 10 in LL29. Network analysis (STRING) revealed enrichment in pathways related to cell adhesion, integrin binding, and hematopoietic cell lineage, highlighting their role in fibroblast activation and ECM remodeling [15].

Proteomic profiling of primary fibroblasts isolated from IPF patients has provided detailed information on disease protein alterations that are potentially lost in non-disease or immortal models. Comparative proteomics of primary fibroblasts from IPF and systemic sclerosis (SSc) patients, using LC-MS combined with Tandem Mass Tag (TMT) labeling, revealed a shared ECM (matrisome) signature with overexpression of proteins such as PLOD2, LUM, POSTN, IGFBP5, GREM1, and SPARC [55], showing similar “fibrotic signatures” for IPF and SSc of activated myofibroblasts, mirroring the results of transcriptomic and miRNA data. Hence, the matrisome proteomic profiles in IPF and SSc lung fibroblasts, together with analyses of mRNA and miRNA, make a crucial contribution to the comprehensive analysis of fibrotic signatures at both the gene and protein levels of IPF and SSc.

In a mouse model of IPF, LC-MS/MS and iTRAQ were applied to examine proteomic changes in primary murine fibroblasts following bleomycin-induced injury. In their study, Della Latta et al. identified key proteins driving fibrotic transformation, including MMPs, OPN, CHI3L1, and CD44, with significant differences in both soluble protein secretion and extracellular vesicle (EV)-derived cargo compared to controls [56] (Figure 5).

In parallel with cell and tissue-based proteomic approaches, extracellular vesicle (EV) proteomics has also become a useful technology for studying intercellular communication in IPF. Mass spectrometry-based proteomic profiling has recently been applied to IPF-derived fibroblast-secreted EVs for the identification and characterization of their protein cargo within the IPF lung microenvironment [57]. This approach, which avoids invasive procedures, may be valuable for both diagnostic and prognostic applications while shedding light on disease-driving mechanisms. The first shotgun proteomics study of EVs isolated from BALF of IPF patients characterized the proteome of the vesicular component of BALF. To achieve this, LC-MS/MS was performed on the total proteins isolated from EVs BALF compared to that of whole BALF in IPF patients, showing considerable differences between them. EVs were enriched in proteins involved in cytoskeleton remodeling, adenosine signaling, adrenergic signaling, C-peptide signaling, and lipid metabolism. These findings support the application of high-sensitivity mass spectrometry to investigate low-abundance, disease-relevant pathways [58]. Moreover, LC-MS/MS of plasma EVs in patients with IPF, chronic hypersensitivity pneumonitis, nonspecific interstitial pneumonitis, and healthy subjects led to the identification of a five-protein biomarker panel, validated via ELISA in an independent cohort, offering potential for differential diagnosis [59]. In another study, label-free proteomic analysis of BALF-derived EVs from bleomycin-treated mice identified 107 proteins enriched in fibrotic vesicles compared to controls. Integrative omics analysis revealed fibroblasts as a major cellular source of BALF-EV cargo, which was enriched in secreted frizzled-related protein 1 (SFRP1), leading to increased transitional cell markers, such as keratin 8, and WNT/β-catenin signaling in primary alveolar type 2 cells. Notably, fibroblast-derived EVs lacking SFRP1 were associated with attenuated fibrosis in vivo, suggesting that SFRP1-EVs function as both biomarkers and potential therapeutic targets [60]. The biomarkers identified in cell lines, primary fibroblasts, mouse models, and EVs are summarized in Figure 5. While tissue- and cell-based proteomic studies offer deep mechanistic insights and identify promising IPF biomarkers, most are based on small cohorts, often limited to single-center, exploratory designs. Therefore, despite high discovery potential, further validation through standardized, longitudinal, and multicenter studies is essential for clinical translation.

### New Insights in Biomarker Discovery with Single-Cell Proteomics

Currently, several antibody-based approaches for quantifying proteins at the single-cell level have been developed, including cytometry by time of flight (CyTOF) [61], single-cell Western blotting [62], and multiplexed protein quantification using immunoassay–PCR coupling [63]. These techniques are able to quantify a few dozen endogenous proteins recognized by specific antibodies. Of particular interest is the application of single-cell proteomics, an emerging frontier that enables quantitative proteome profiling at near-single-cell resolution, aiming at a multi-layered characterization of the proteome signature compared to bulk MS. By capturing protein-level heterogeneity, single-cell proteomics provides valuable insights into cellular diversity, functional specialization, and pathological reprogramming, although current studies remain largely focused on expression-level proteomics rather than post-translational modifications.

Single-cell proteomics poses significant technical challenges due to extremely low amounts of protein present in a single cell [6,7,64], coupled with the fact that proteins cannot be amplified as nucleic acids can, despite protein levels typically exceeding transcript abundance by more than two orders of magnitude. Nowadays, single-cell proteomics technology is based on advances in sample preparation, MS instrumentation, and data acquisition that lead to an ultrasensitive single-cell resolution. Specifically, it consists of lysis and proteolytic digestion of a limited number of cells in nanoliter droplets, followed by direct loading onto the analytical column [65].

A representative study employing the Single-Cell Proteomics by Mass Spectrometry (SCoPE) approach demonstrated the potential of this method: individually isolated live cells were lysed by sonication, digested with trypsin, and TMT-labeled peptides were combined and analyzed via LC–MS/MS, resulting in the quantification of approximately 1000 proteins across 100 single tumor cells [66]. These data underscore the capacity of modern MS-based platforms to achieve deep proteome coverage from extremely low sample inputs [66].

At present, single-cell proteomics remains an evolving methodological platform and has not yet been applied directly to IPF lung tissue at a true single-cell resolution. Nonetheless, the use of robust scRNA-seq and spatial proteomics provides highly relevant insights into the pathological mechanisms and target cell populations in IPF lungs. Then, integrating bulk tissue proteomics, single-cell platforms, EV profiling, and spatial proteomics helps to delineate layered protein dysregulation and paracrine signaling networks that remain undetectable through transcriptomics alone. Ultimately, these emerging technologies are transforming the conceptual understanding of IPF from a generalized fibrotic response into a spatially organized and functionally specialized process, paving the way for the identification of cell-type-specific biomarkers and precision therapeutic targets.

## 8. Key Validated Protein Biomarkers in IPF

Among the different protein biomarkers that have been investigated in IPF, only a few have shown consistent validation across independent studies. In this section, we highlight the most robust and frequently reported biomarkers, summarizing their biological roles, diagnostic and prognostic utility, and limitations (Table 1). According to the literature, biomarkers such as MMP-7, KL-6, SP-D, and CCL18, have demonstrated repeated associations with IPF pathogenesis across multiple studies and patient populations [67,68,69]. These biomarkers were linked to crucial pathophysiological features of IPF, such as epithelial injury, ECM remodeling, and inflammatory signaling, corroborating their central role in disease biology. MMP-7 was associated with disease severity and progression in eight independent studies, including large multicenter cohorts [30,70], although its diagnostic specificity remains limited due to its expression in other lung diseases, raising concerns about diagnostic specificity [71]. KL-6, a high-molecular-weight glycoprotein expressed on regenerating alveolar type II cells, was linked to worse prognosis and disease progression in cohorts exceeding 300 patients, but it lacks standardized measurement thresholds across centers [33].

Among these biomarkers, periostin, a matricellular protein associated with fibroblast activation, has been less frequently studied and is currently considered an exploratory biomarker, potentially relevant during acute exacerbations or in fibrotic subphenotypes with active remodeling. Periostin remains an exploratory biomarker in the context of IPF, primarily due to the limited number of large, multicenter, and prospective validation studies compared to more established biomarkers, such as MMP-7, KL-6, or SP-D. For example, the study by Naik P.K. et al. demonstrated an association between periostin levels and disease progression in IPF patients, but it included only 54 individuals with approximately 48 weeks of follow-up [72]. A subsequent multicenter study investigating monomeric periostin assessed its diagnostic and short-term prognostic potential in 60 IPF patients and 137 healthy controls [73]. Another major limitation is the lack of assay standardization and poor reproducibility across studies. Various studies have used different analytical platforms to measure either total or monomeric periostin, leading to inconsistent results and complicating direct comparisons. Additionally, differences in follow-up durations and the absence of standardized clinical thresholds further hinder its integration into routine clinical practice. Similarly, CCL18, despite its repeated association with poor prognosis, still requires broader validation across ethnically and clinically diverse populations. Thus, MMP-7, KL-6, and SP-D are the most reliable and well-supported biomarkers in IPF, demonstrating consistent diagnostic or prognostic utility across multiple studies and contributing significantly to treatment follow-up, while periostin or CCL18, though promising, require further multicenter validation to assess their specificity and clinical applicability [33] (Table 1). Finally, most biomarkers show greater utility when used in panels rather than alone, due to overlapping profiles with other ILDs. For this purpose, the systematic review by Rezaeeyan et al. [74], which analyzed protein biomarkers identified through proteomics across five various respiratory diseases, including IPF, asthma, COPD, obliterative bronchiolitis, and chemical warfare victims exposed to mustard gas, reported 5638 proteins altered in IPF-related studies. Finally, the 31 proteins that were commonly expressed in all five diseases regulate different factors and molecular pathways, leading either to the regulation of inflammatory pathways, in the case of COPD and mustard-exposed patients, or lung fibrosis, in the case of IPF. Thus, while MMP-7, KL-6, and SP-D have demonstrated diagnostic and prognostic potential in IPF, their clinical translation remains incomplete. Finally, integrating these biomarkers into diagnostic and therapeutic pathways could enhance early detection, disease stratification, and personalized treatment strategies. For example, KL-6 and SP-D have been proposed as tools to support clinical practice in some countries (e.g., Japan, China) [75] for the differential diagnosis of IPF versus other ILDs, especially when combined with radiological and functional assessments.

**Table 1 biomedicines-13-02656-t001:** Key validated protein biomarkers in IPF and their biological role, strengths, and limitations.

Biomarker	Biological Role/Source	Diagnostic/Prognostic Value	Strengths	Limitations	References
MMP-7 (Matrix Metalloproteinase-7)	ECM remodeling enzyme secreted by alveolar epithelium and macrophages	Elevated in IPF; baseline levels predict poor survival	Strong predictor of progression; frequently validated; independent in multivariate models	Not IPF-specific; elevated in other ILDs; cutoff variability	[68,76]
KL-6 (Krebs von den Lungen-6/MUC1)	Marker of type II pneumocyte injury	High in IPF; inversely correlates with FVC and DLCO; associated with disease progression	Highly sensitive; useful for monitoring	Limited specificity; variation across studies and platforms	[43,67,77,78,79,80]
Surfactant proteins A and D (SP-A, SP-D)	Alveolar epithelial cell products involved in surfactant metabolism and immune response	Increased in IPF; especially SP-D correlates with disease severity and outcomes	Well established in serum and BAL; measurable in standard biofluids	SP-A less consistent; levels influenced by inflammation	[39]
CCL18 (Chemokine [C-C motif] ligand 18)	Produced by alveolar macrophages; involved in immune cell recruitment and collagen production	Elevated in IPF; predictive of FVC decline and survival	Promising prognostic marker; measurable in serum	Lacks disease specificity; varies across cohorts	[33,41,81,82,83,84]
Periostin	ECM protein secreted by activated fibroblasts; marker of fibrosis and tissue remodeling	Elevated in some IPF cohorts; may correlate with progression or acute exacerbations	Novel marker; may reflect fibroblast activity	Fewer studies; requires further validation	[85,86,87,88]

## 9. AI-Powered Proteomics: A New Era in Translational Medicine

In complex, high-dimensional datasets of mass spectrometry-based proteomics, Artificial Intelligence (AI), particularly machine learning and deep learning, can improve peptide and protein identification, thereby enhancing quantification accuracy. AI-driven computational and predictive models enhance proteomic workflows for biomarker discovery, identifying protein expression profiles that specifically discriminate between healthy and disease states in various medical fields, such as cardiovascular diseases, cancer, and neurodegenerative diseases [89,90]. Deep learning can predict experimental peptide measurements from amino acid sequences across large proteomic datasets, resulting in greater reliability and consistency in MS workflows. Moreover, an AI technology named federated deep learning (e.g., ProCanFDL) allows the secure sharing and analysis of large datasets across several institutions without sharing confidential patient data [91]. AI also facilitates the integration of proteomics with other omics techniques, such as genomics, transcriptomics, and metabolomics, which is essential for identifying novel therapeutic targets and developing personalized treatment strategies [89].

AI and machine learning applied to the analysis of mass spectrometry-derived proteomic data have been applied in IPF research for advancement in both biomarker identification and disease classification. To date, serum proteomics analyzed by DIA, together with machine learning, has stratified IPF patients into distinct molecular subgroups associated with clinical outcomes, identifying age-related pathways and proteins such as LDHA and CCT6A. A machine-learned combinatorial biomarker panel successfully distinguished IPF patients from healthy individuals [92]. Furthermore, plasma proteomic data from approximately 1600 patients with IPF or connective tissue disease-associated interstitial lung disease were analyzed using recursive feature elimination and multiple classifiers, such as support vector machine (SVM) and Least Absolute Shrinkage and Selection Operator (LASSO), which are suitable and reliable for high-dimensional proteomic data. This resulted in a 37-protein classifier (PC37) that, with high sensitivity and specificity, was able to distinguish IPF from other ILDs [93]. Additionally, integrated analyses of lung transcriptomic and plasma proteomic datasets supported by an AI-based analytic approach identified a set of 34 differentially expressed analytes in IPF samples compared to healthy controls. IPF samples showed strong enrichment in chemotaxis, tumor infiltration, and mast cell migration pathways as well as downregulated ECM degradation together with upregulation of mucosal (CCL25 and CCL28) and Th2 (CCL17 and CCL22) chemokines—all highly correlated within subjects [94]. The direct application of deep learning models to raw MS spectra—for proteomics—offers the potential for automated feature discovery and improved interpretability in IPF proteomic research, thus facilitating translation to the clinic. Despite its growing potential, the use of AI in proteomics—particularly in clinical settings such as IPF—faces significant limitations. One of the primary concerns is reproducibility: AI and machine learning models trained on limited or institution-specific datasets often struggle to generalize across diverse populations due to variability in sample preparation, instrumentation, and clinical heterogeneity. Furthermore, AI models can be highly sensitive to missing values and batch effects—issues common in proteomic studies of rare diseases such as IPF. Without rigorous external and multicenter validation, results may not translate beyond the original study cohort, limiting clinical utility.

Finally, the integration of AI into healthcare systems will also require substantial infrastructural, financial, and regulatory investment.

## 10. Clinical Translation of Proteomics from Bench to Bedside in IPF

Biological markers identified through mass spectrometry are essential for predicting the course and outcome of IPF.

In general, biomarkers, when paired with medical evaluations, allow for the creation of predictive models that estimate individual risks related to disease progression and mortality [26]. IPF patients display variable rates of disease progression and treatment response, making the condition inherently heterogeneous. Clinicians can use biomarkers to detect progressive phenotypes, which is essential for tailoring therapeutic interventions and disease management to each patient’s needs [5]. To strengthen clinical relevance, it is essential to contextualize how specific proteomic biomarkers can be realistically integrated into patient care. MMP-7 and SP-D, when combined with clinical parameters and imaging, have shown potential to discriminate IPF from other ILDs, such as chronic hypersensitivity pneumonitis and nonspecific interstitial pneumonia, where radiological findings often overlap [71]. Furthermore, baseline KL-6 and periostin levels have been associated with response to antifibrotic therapy (nintedanib and pirfenidone), suggesting their use in guiding treatment decisions and monitoring efficacy over time [95]. In addition, elevated levels of CCL18 and surfactant protein D have been associated with a higher risk of acute exacerbations, which represents one of the most severe complications of IPF. Integrating these biomarkers into clinical models could enable risk stratification and personalized management strategies [26]. By integrating these biomarkers into multiparametric models that include radiologic, genomic, and functional data, clinicians may better predict disease trajectory, individualize follow-up intensity, and optimize timing for interventions such as transplant referral or clinical trial enrollment. These practical scenarios highlight the true translational potential of proteomics beyond theoretical promise.

To guarantee scalability and feasibility for the clinical application of biomarkers and to improve patient prognosis in IPF, collaboration between academic, clinical, and industry partners is essential, as is adherence to regulatory guidelines. The use of biomarkers in IPF management holds a promising future, with the introduction of new technologies in the fields of single-cell sequencing and proteomics, genomic profiling, and spatial transcriptomics and proteomics, which will deepen our understanding of the disease and help identify novel biomarkers.

### Current Limitations in the Clinical Translation of Proteomics

Cost remains a significant issue for the widespread use of proteomics in clinical protocols. Thus, it is critical to advance technologies that are both effective and economically sustainable with the implementation of biochemical protocols that allow high-throughput analyses of multiple samples adapted to the requirements of routine clinical practice. Another concern relates to the limited awareness and integration of mass spectrometry technologies within the clinical community, as their use has largely remained confined to basic and preclinical research settings, rather than certified clinical diagnostic laboratories, where formal validation is mandatory prior to clinical implementation in the routine patient care [96].

Thus, the transition of LC-MS techniques from research laboratories into clinical settings requires compliance with Clinical Laboratory Improvement Amendments (CLIA) standards, which imply adequate laboratory space and infrastructure, staff training, competence, and quality management [97]. For this reason, only a limited number of FDA-approved diagnostic assays, such as MALDI-TOF MS, are currently available. Nowadays, MS-based diagnostic tests and devices that measure or identify biomarkers are rarely applied in clinical practice [98], while only a few biomarkers identified through mass spectrometry and validated with orthogonal assays in one or several independent cohorts have been successfully translated into clinical settings [99,100]. Then, if a panel of different biomarkers needs to be validated, a subsequent computational model based on AI [101] must be employed for precise quantification, which adds further costs and requires additional staff training and expertise.

## 11. Conclusions

Different MS approaches—from GC-MS analysis of volatile organic compounds in exhaled breath to innovative LC-MS/MS applications on plasma, bronchoalveolar lavage fluid, lung tissue, cell line, primary cells, and extracellular vesicles—have uncovered a broad spectrum of IPF biomarkers associated with crucial molecular pathways such as ECM remodeling, immune regulation, oxidative stress, inflammation, and fibrotic signaling (Table 2). Furthermore, several key proteins, such as MMP-7, KL-6, and SP-D, as summarized in Table 1, have consistently emerged across multiple sample types and studies, underscoring their potential as reliable biomarkers of disease progression and therapeutic targets, while periostin and CCL18 remain promising candidates requiring further multicenter validation to confirm their clinical applicability. Hence, focusing on biomarkers consistently replicated across multiple studies, the application of spatial proteomics and single-cell techniques—enhancing resolution at both tissue and cellular levels—will provide novel insights into their role in IPF pathogenesis and new therapeutic approaches for personalized care in IPF.

Taken together, these findings reveal the potential of proteomic mass spectrometry to deepen our understanding of IPF, facilitating the development of diagnostic and prognostic tools as well as new targeted therapies.

## Figures and Tables

**Figure 1 biomedicines-13-02656-f001:**
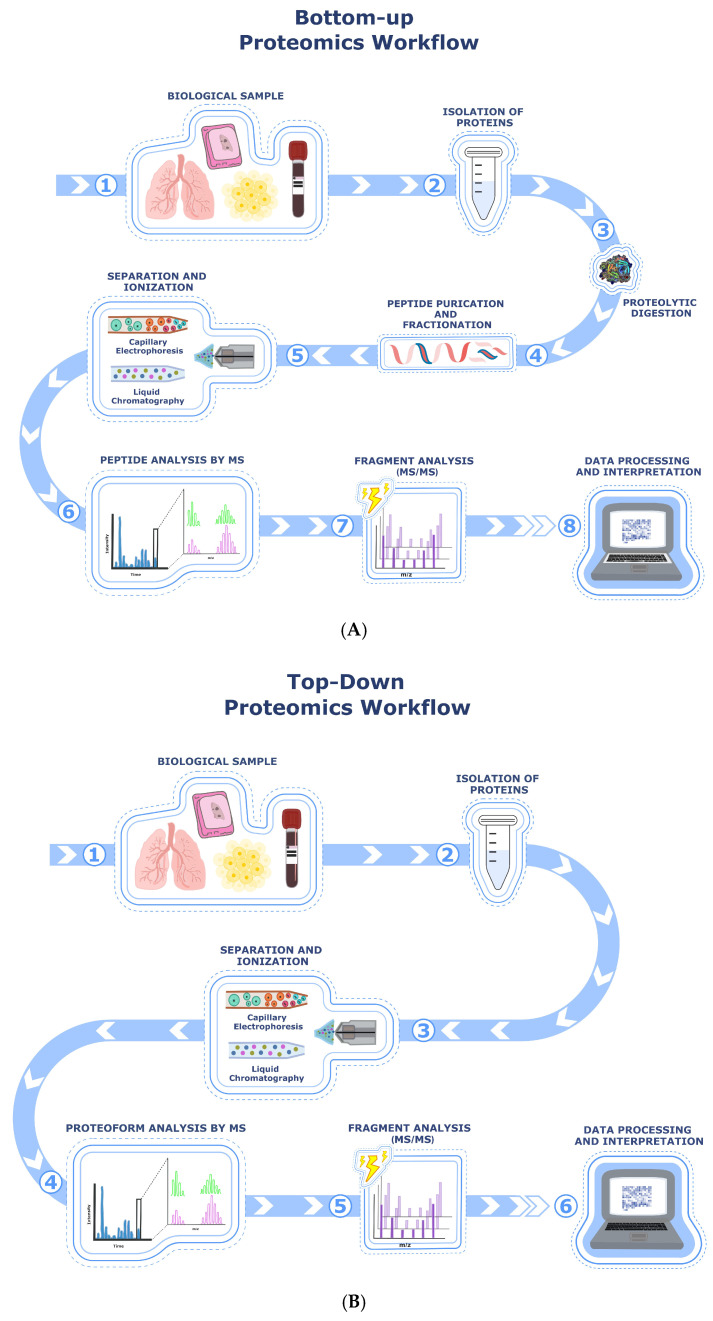
(**A**) Bottom-up proteomics is based on the indirect identification of proteins through their constituent peptides. The workflow starts with the acquisition of a biological sample (1) and the extraction of total proteins (2), followed by enzymatic digestion, commonly with trypsin, to cleave proteins at defined sites and generate peptides suitable for MS analysis (3). Peptides are then purified and, when necessary, fractionated using techniques such as strong cation exchange (SCX), high-pH reversed-phase liquid chromatography, or other orthogonal methods to reduce sample complexity (4). This is followed by peptide separation using nanoLC or capillary electrophoresis and ionization by electrospray (5), enabling MS1 analysis for peptide profiling and quantification (6). Selected precursor ions undergo fragmentation, typically via higher-energy collisional dissociation (HCD) or collision-induced dissociation (CID), to produce fragment ion spectra (7), which are analyzed by tandem mass spectrometry (MS/MS) to determine peptide sequences (8). The resulting spectra are interpreted using database search engines and bioinformatics pipelines to identify peptides and infer the presence, abundance, and post-translational modifications (PTMs) of the corresponding proteins. Although widely applicable and highly scalable, this approach does not preserve information at the proteoform level and may obscure isoform-specific and PTM-specific distinctions. (**B**) Top-down proteomics enables the direct analysis of intact proteoforms without enzymatic digestion. The workflow starts with the acquisition of a biological sample (1), followed by the extraction of intact proteins under native or denaturing conditions (2). After protein separation, typically via reversed-phase liquid chromatography (RPLC) or capillary electrophoresis (CE), and electrospray ionization (3), intact protein ions are introduced into the mass spectrometer for high-resolution proteoform analysis (4). Gas-phase fragmentation techniques, such as electron capture dissociation (ECD), electron transfer dissociation (ETD), or higher-energy collisional dissociation (HCD), are then applied to generate fragment ions (5), which are analyzed by tandem MS to obtain sequence information (6). Computational tools subsequently perform spectral deconvolution, proteoform identification, and localization of post-translational modifications (PTMs) and sequence variants. This approach provides direct, site-resolved characterization of proteoforms, although it is currently limited by dynamic range and overall proteome coverage.

**Figure 2 biomedicines-13-02656-f002:**
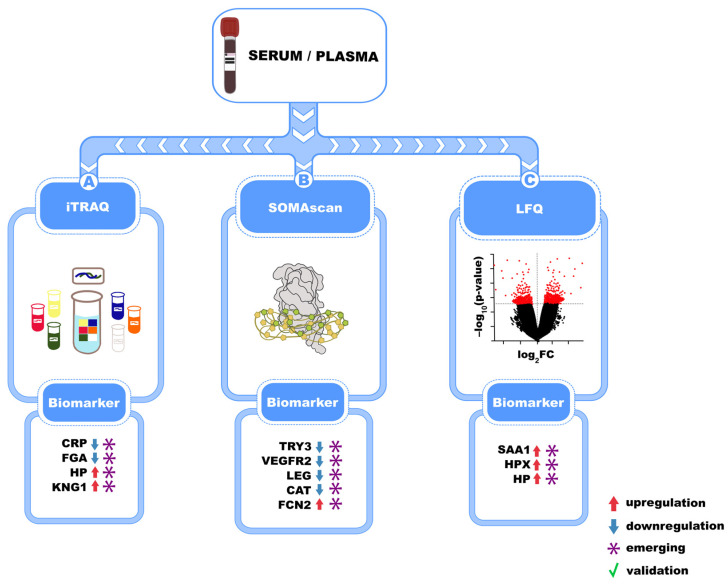
Schematic representation of mass spectrometry-based approaches to discovering serum and plasma biomarkers in idiopathic pulmonary fibrosis (IPF). iTRAQ, SOMAscan, and label-free quantification (LFQ) technologies have enabled the identification of several circulating protein biomarkers, including CRP, HP, VEGFR2, SAA1, and FCN2, which are shown as either up-regulated or down-regulated according to the figure legend. Some of these are emerging, exploratory biomarkers (*), while others are validated biomarkers (green checkmark).

**Figure 3 biomedicines-13-02656-f003:**
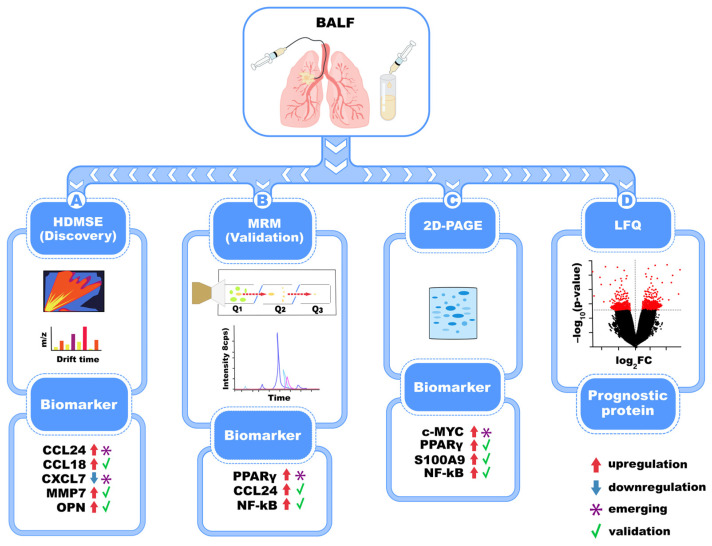
Proteomic analysis of BALF detects several candidate biomarkers involved in IPF pathogenesis. HDMSE, MRM, 2D-PAGE, and LFQ detect inflammatory and fibrotic proteins, like MMP7, CCL18, CXCL7, CCL24, and S100A9, and dysregulated pathways, like NF-κB, PPARγ, and c-MYC which are shown as either up-regulated or down-regulated according to the figure legend. Some of these are emerging, exploratory biomarkers (*), while others are validated biomarkers (green checkmark).

**Figure 4 biomedicines-13-02656-f004:**
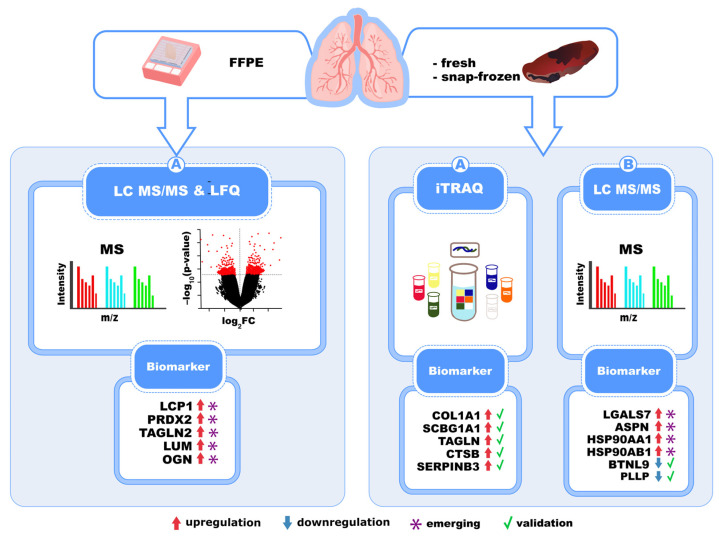
Proteomic investigation of lung tissue from IPF patient samples reveals key biomarkers and fibrogenesis pathways. Protein analyses with LC-MS/MS, LFQ, and iTRAQ show upregulation of LCP1, PRDX2, TAGLN2, LUM, OGN, and ECM-associated and stress-response proteins, such as COL1A1, SCGB1A1, HSP90AA1, and HSP90AB1 (red arrows). Downregulated proteins BTNL9 and PLLP are implicated in epithelial differentiation and protective roles against fibrosis (blue arrows). Some of these are emerging, exploratory biomarkers (*), while others are validated biomarkers (green checkmark).

**Figure 5 biomedicines-13-02656-f005:**
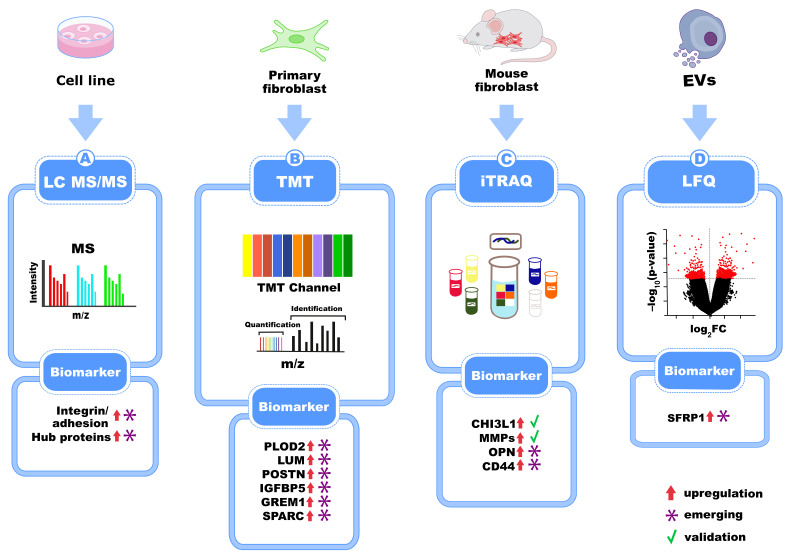
Overview of proteomic studies on IPF patient-derived pulmonary cells, cell lines, and mouse models. MS-based approaches like LC-MS/MS, TMT, iTRAQ, and LFQ have identified major matrisome proteins (PLOD2, LUM, POSTN), fibrotic mediators (IGFBP5, MMPs, OPN), and secreted factors via extracellular vesicles (SFRP1, ANXA5) which are shown up-regulated according to the figure legend. Some of these are emerging, exploratory biomarkers (*), while others are validated biomarkers (green checkmark).

**Table 2 biomedicines-13-02656-t002:** Mass spectrometry-based biomarker discovery in IPF and related pathways.

Mass Spectrometry Technique	Identified Biomarkers	Activated Molecular Pathway	Sample Type	Reference(s)
GC-MS	Acetone, other VOCs	Oxidative stress, inflammation	Exhaled breath (VOCs)	[23]
LC-MS/MS	MMP-7, MMP-1, MMP-10	ECM remodeling	Plasma/serum	[34,35,36,37,38]
LC-MS/MS	CCL18	Inflammation, immunoregulation	Plasma/serum	[39,40]
LC-MS/MS	Periostin	TGF-β, IL-4, IL-13 signaling	Plasma/serum	[41,42,43]
LC-MS/MS	SP-A, SP-D, KL-6	Immune response, alveolar regeneration	Serum	[33,40,44]
SOMAscan (aptamer-based)	VEGFR2, Ficolin-2, Legumain, Cathepsin, ICOS, Trypsin-3	IPF progression (not specified)	Plasma	[45]
iTRAQ + LC-MS/MS	CRP, Fibrinogen-α, Haptoglobin, Kininogen-1	Systemic inflammation	Plasma	[46]
Quantitative proteomics	SAA1, Haptoglobin, Hemopexin	Inflammation, oxidative stress, ECM	Plasma	[47]
HDMSE, MRM, 2D-PAGE, MALDI-ToF	MMP-7, CXCL7, CCL18, S100A9, ILs, MIF, Calgranulin B, CCL24	Inflammation, ECM remodeling, cellular signaling	BALF	[48,49,50,51,52]
LC-MS/MS (FFPE tissue)	LCP1, PRDX2, TAGLN2, LUM, OGN	TGF-β, cell adhesion, ECM	Lung tissue (FFPE)	[15]
iTRAQ + LC-MS/MS	COL1A1, SCGB1A1, HSP90AA1/AB1, LGALS7, ASPN	ECM production and remodeling	Fresh lung tissue	[55]
LCM-MS, spatial proteomics	TGF-β1/2/3, LTBP1, FN1, SFRP1	TGF-β signaling, ECM remodeling, EMT	Laser-captured lung tissue sections	[57,58,59,60]
Label-free LC-MS/MS	>80 proteins (e.g., POSTN, IGFBP5, SPARC)	Matrisome, cell adhesion, ECM signaling	Primary cell lines (fibroblasts)	[61,62]
LC-MS/MS (EVs from BALF and plasma)	SFRP1, signaling proteins, cytokines, cytoskeletal proteins	WNT/β-catenin, cell–cell communication	EVs from BALF and plasma	[63,64,65,66,67]

## Data Availability

No new data were created or analyzed in this study. Data sharing is not applicable to this article.

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
