# Peer review of "Predicting and Treating Pulmonary Fibrosis with Proteomic Biomarker Investigations"

_biomedicines, 2025, doi:10.3390/biomedicines13112656_

Round 1

Reviewer 1 Report

Comments and Suggestions for Authors

The review entitled "Predicting and treating pulmonary fibrosis with proteomic biomarker investigations" covers an important and timely topic, focusing on the use of proteomics to identify biomarkers in idiopathic pulmonary fibrosis (IPF). The manuscript is broad and mostly up to date, looking at different sample types, technologies, and translational perspectives, and I think it will be of interest to readers of Biomedicines. That said, the paper is sometimes very descriptive, and at times it can be a bit hard to follow. A slightly clearer structure, a more critical synthesis, and some careful language polishing would improve its overall impact.

The introduction spends a lot of space on general definitions of omics technologies. I would suggest shortening this part and concentrating more on why proteomics is particularly relevant for IPF.

Many biomarkers are mentioned throughout the manuscript, but they are not always discussed in detail. It could be helpful to highlight a shorter list of biomarkers that have been consistently validated, for instance MMP-7, CCL18, KL-6, periostin, and surfactant proteins, and to point out their strengths and limitations. Right now, the large number of details and the lack of a clear classification can make it hard to grasp the key points. It might also be useful to briefly compare these findings with previous literature—for example, Rezaeeyan et al., 2023 (Immun Inflamm Dis, 11: e1090), which reviewed protein biomarkers in respiratory diseases using proteomic approaches. Adding this kind of comparison would give readers a better sense of which biomarkers are well-supported and which are still exploratory.

Figures and tables could be presented more effectively. Table 1 is informative but could be reorganized to highlight biomarkers that have been independently validated. Figure 1 is quite dense and a bit confusing, as it doesn’t clearly show that protein analysis by MS, fragmentation, and fragment analysis (MS/MS) are part of the same workflow. Emphasizing the main biomarkers, breaking the text into smaller subsections, and adding targeted figures or summary tables would make the paper easier to read.

It could also be useful to include a dedicated section or figure summarizing the key take-home messages. This could be a short list or a simple schematic showing the most consistently validated biomarkers, their strengths and limitations, and the potential of emerging technologies. This kind of summary would make the main points more accessible.

The conclusion could be tighter. Focusing on the few biomarkers that are consistently replicated and mentioning the promise of new technologies, such as spatial proteomics, single-cell approaches, and AI, would make the take-home message clearer.

Comments on the Quality of English Language

The language is generally understandable, but there are some typographical errors and awkward expressions that could be corrected. For instance: “UIP pa ern” (p. 2, l. 51 → pattern), “bo om-up” (p. 5, l. 196 → bottom-up), “se ings” (p. 14, l. 619 → settings), “poor survival times” (p. 2, l. 60 → poor survival rates), “enabled to enter the era” (p. 2, ll. 84–85 → ushered in the era), or “a proven diagnosis… may represent a challenge” (p. 3, l. 104 → establishing a definitive diagnosis remains challenging). Some sentences are quite long and could be split to improve clarity. Standardizing acronyms (e.g., UIP, BALF, AE, PTMs) would also help. 

Author Response

On behalf of all the Authors we would like to thank all the Reviewers for their valuable time and useful contribution to our Manuscript Review “Predicting and treating pulmonary fibrosis with proteomic biomarker investigations”. We strongly appreciate the suggestions and inputs received that will definitely improve our manuscript Review with clearer structure and a more critical synthesis, improving its overall impact. Thus, following the Reviewers suggestions, we have also reorganized the manuscript improving and integrating the Figures. Please find below the detailed responses to Reviewers.

Reviewer 1

The review entitled "Predicting and treating pulmonary fibrosis with proteomic biomarker investigations" covers an important and timely topic, focusing on the use of proteomics to identify biomarkers in idiopathic pulmonary fibrosis (IPF). The manuscript is broad and mostly up to date, looking at different sample types, technologies, and translational perspectives, and I think it will be of interest to readers of Biomedicines. That said, the paper is sometimes very descriptive, and at times it can be a bit hard to follow. A slightly clearer structure, a more critical synthesis, and some careful language polishing would improve its overall impact. The introduction spends a lot of space on general definitions of omics technologies. I would suggest shortening this part and concentrating more on why proteomics is particularly relevant for IPF.

We thank the Reviewer for this valuable suggestion. In the revised manuscript, we have significantly shortened the general description of omics technologies in the Introduction section to improve focus and clarity. We also expanded the discussion on the specific relevance of proteomics in IPF, particularly highlighting its unique ability to capture post-translational modifications and dynamic protein-level changes that are not accessible through genomics or transcriptomics. These aspects make proteomics especially suited to investigate the complex pathophysiological processes and acute exacerbations associated with IPF. The revised paragraph can be found on page 2 Line 69-108.

“Many biomarkers are mentioned throughout the manuscript, but they are not always discussed in detail. It could be helpful to highlight a shorter list of biomarkers that have been consistently validated, for instance MMP-7, CCL18, KL-6, periostin, and surfactant proteins, and to point out their strengths and limitations. Right now, the large number of details and the lack of a clear classification can make it hard to grasp the key points. It might also be useful to briefly compare these findings with previous literature—for example, Rezaeeyan et al., 2023 (Immun Inflamm Dis, 11: e1090), which reviewed protein biomarkers in respiratory diseases using proteomic approaches. Adding this kind of comparison would give readers a better sense of which biomarkers are well-supported and which are still exploratory.”

We thank the Reviewer for this constructive comment. In response, we have revised and integrated the section on IPF-related protein biomarkers with a specific section (8 Key validated Protein Biomarkers in IPF, pag 17 Line 671-700) to improve clarity and focus. Specifically, we now highlight a concise list, organized in Table 1 (pag 17-18), of consistently validated biomarkers, including MMP-7, CCL18, KL-6, periostin, and surfactant proteins (SP-A, SP-D) that higlights their biological relevance, strengths, and limitations in the context of diagnosis and prognosis of IPF, clarifying which biomarkers show the strongest clinical evidence versus those that remain exploratory and need further validation. Then, following the Reviewer suggestions to further enhance the impact of this section, we added a comparative summary referencing the review by Rezaeeyan et al., 2023, which analyzed proteomics-based biomarkers across various respiratory diseases (page 17 Line 696-700).

Figures and tables could be presented more effectively. Table 1 is informative but could be reorganized to highlight biomarkers that have been independently validated. Figure 1 is quite dense and a bit confusing, as it doesn’t clearly show that protein analysis by MS, fragmentation, and fragment analysis (MS/MS) are part of the same workflow. Emphasizing the main biomarkers, breaking the text into smaller subsections, and adding targeted figures or summary tables would make the paper easier to read. It could also be useful to include a dedicated section or figure summarizing the key take-home messages. This could be a short list or a simple schematic showing the most consistently validated biomarkers, their strengths and limitations, and the potential of emerging technologies. This kind of summary would make the main points more accessible. The conclusion could be tighter. Focusing on the few biomarkers that are consistently replicated and mentioning the promise of new technologies, such as spatial proteomics, single-cell approaches, and AI, would make the take-home message clearer.

We thank the Reviewer for this valuable feedback and following this suggestion we added the Table 1 whose details have been explained in the previous response (Pag 18). In particular Table 1 has been added to clearly distinguish between biomarkers that have been independently validated across multiple studies and those that remain exploratory. We believe that this reorganization improves the clarity and relevance of the table content for readers. Table 1 in the first version of the Manuscript became Table 2 (Pag. 21-22) where we wanted to list the whole biomarkers related to IPF identified through proteomic studies pointing out the techniques crucial for their identification and their related pathways. Figure 1 has been redesigned to better represent the complete mass spectrometry workflow. The updated version now clearly illustrates the sequential steps of protein extraction, digestion, MS analysis, peptide fragmentation (MS/MS), and data interpretation as part of a single, coherent process. In addition, we have created new targeted figures that highlight bot exploratory and validated biomarkers identified through proteomics from different biological sources, including Serum Plasma (Figure 2, Pag 10) BALF (Figure 3, Pag. 12) lung tissue (Figure 4, Pag.13) and cell line and primary cells (Figure 5, Pag 16). These visualizations aim to improve readability and help the reader easily understand which biomarkers are associated with each sample type.

Furthermore, following the suggestions of the Reviewer 1, we have revised the conclusion section to make it more concise and focused. The updated version emphasizes the most consistently validated protein biomarkers (such as MMP-7, KL-6, and SP-D) (Pag 21 Line 837-859), and it highlights their clinical relevance, and reflects on the emerging potential of spatial proteomics, single-cell technologies, and AI-driven analyses in reshaping biomarker discovery and personalized care in IPF (Pag 21 Line 841-846). With these integrations we hope to deliver a clearer and more impactful take-home message for readers.

The language is generally understandable, but there are some typographical errors and awkward expressions that could be corrected. For instance: “UIP pa ern” (p. 2, l. 51 → pattern), “bo om-up” (p. 5, l. 196 → bottom-up), “se ings” (p. 14, l. 619 → settings), “poor survival times” (p. 2, l. 60 → poor survival rates), “enabled to enter the era” (p. 2, ll. 84–85 → ushered in the era), or “a proven diagnosis… may represent a challenge” (p. 3, l. 104 → establishing a definitive diagnosis remains challenging). Some sentences are quite long and could be split to improve clarity. Standardizing acronyms (e.g., UIP, BALF, AE, PTMs) would also help.

We thank the Reviewer for the detailed and constructive feedback regarding the language. We have carefully revised the manuscript to correct all identified typographical errors that the Reviewer mentioned, however we were not able to find them in our latest version submitted to the journal (e.g., “UIP pa ern” → “UIP pattern”, “bo om-up” → “bottom-up”, “se ings” → “settings”). Nonetheless, we have successfully improved the following expressions as suggested by the Reviewer: “enabled to enter the era” was deleted in the updated shortest version of the Introduction (p. 3, Track-changes erased lines 97–98), and “a proven diagnosis may represent a challenge” has been substituted with → “establishing a definitive diagnosis remains challenging” (p. 3, l. 120-121).

Reviewer 2 Report

Comments and Suggestions for Authors

The article is a topical issue as idiopathic pulmonary fibrosis (IPF) remains a disease with bad prognosis and limited therapeutic option. As much as the authors provide a broad overview of proteomic tools and candidate biomarkers, the paper in its current state contains significant shortcomings related to focus, depth of analysis, and clinical utility. In present, the article is as list-like a description of proteomics techniques as it can yet become a translational review article.

An important issue is a deficiency of narrative continuity and coherence. Coverage attempts are made to detail nearly all proteomic methods from GC-MS and LC-MS through MALDI, NMR, single-cell proteomics, spatial proteomics, extracellular vesicles, and machine learning. However, these parts are presented as isolated summaries without any unifying message. Instead of indicating the way to which approaches are most likely to be worthwhile for IPF management, the paper shatters the reader's focus among too numerous technical descriptions. This encyclopedic strategy lessens the clinical problem in hand. More weight should be given to approaches that bear some actual translational promise, and the more peripheral approaches mentioned briefly.

Some biomarkers including MMP-7, CCL18, periostin, KL-6, and surfactant proteins SP-A and SP-D are presented in several sections with almost identical wording. Rather than summarizing information, the authors duplicate it in plasma, BALF, tissue, and cellular subsections. Comparative synthesis is absent: the review could benefit from synthesized tables or figures stratifying biomarkers by validity status, reproducibility, and clinical usefulness, and not listing the same molecules again and again.

Most importantly, the manuscript fails in critical appraisal. Most biomarker studies are reported uncritically, without mentioning sample size, study design, replication, or confounding variables. For instance, while KL-6 and MMP-7 are repeatedly highlighted as the most promising, no explanation is given for why they remain beyond standard clinical care after decades of research. Similarly, periostin and CCL18 are reported to be biomarkers predictive of IPF without illustrating their non-specificity or the difficulty of distinguishing IPF from other interstitial lung diseases.

The clinical translation sections are tenuous. The authors predict proteomics will improve diagnosis, prognosis, and therapy but fail to demonstrate how some biomarkers could realistically be used in everyday clinical practice. Scenarios such as differential diagnosis relative to other ILDs, prediction of response to antifibrotic therapy, or risk stratification for acute exacerbation are not addressed in depth. Without such context, the argument remains in the abstract and fails to convince the reader of clinical application.

The inclusion of artificial intelligence in Section 8 is a weakness. The description of the account is shallow and complacent, portraying AI as a transformative device without calling out the largest dangers of reproducibility, interpretability, heterogeneity of data, and validation. There has to be some balance of discussion here; otherwise, the section reads more like an advertisement than a critical scholarship.

Tables and figures can be done better: Figure 1 is essentially a generic textbook figure of proteomic workflows and not IPF-specific, and Table 1 is exhaustive but intimidating. Neither conveys to the reader at a glance which biomarkers have the greatest potential for translation. A reorganized table by strength of evidence, validation status, and reproducibility would be much more useful.

Comments on the Quality of English Language

There are a few issues of language and style as well. Typos, awkward phrasing, and overly verbose jargon sentences confuse a great deal. The manuscript could use a good close edit by a native English speaker, too.

Author Response

Reviewer 2

The article is a topical issue as idiopathic pulmonary fibrosis (IPF) remains a disease with bad prognosis and limited therapeutic option. As much as the authors provide a broad overview of proteomic tools and candidate biomarkers, the paper in its current state contains significant shortcomings related to focus, depth of analysis, and clinical utility. In present, the article is as list-like a description of proteomics techniques as it can yet become a translational review article.

An important issue is a deficiency of narrative continuity and coherence. Coverage attempts are made to detail nearly all proteomic methods from GC-MS and LC-MS through MALDI, NMR, single-cell proteomics, spatial proteomics, extracellular vesicles, and machine learning. However, these parts are presented as isolated summaries without any unifying message. Instead of indicating the way to which approaches are most likely to be worthwhile for IPF management, the paper shatters the reader's focus among too numerous technical descriptions. This encyclopedic strategy lessens the clinical problem in hand. More weight should be given to approaches that bear some actual translational promise, and the more peripheral approaches mentioned briefly.

We thank the reviewer for this insightful comment. We fully agree that the initial version lacked a coherent narrative thread and may have overwhelmed the reader with technical descriptions. In response, we have revised the manuscript to enhance narrative flow and coherence, starting to shortening the introduction and removing technical details about other omics techniques (Pag 2-3 Line 62-108) to proceed insight the text. Specifically:

  • We restructured the sections 3 now entitled “Protein analysis through different type of mass spectrometry relevant for IPF research” (pages 4-5-6 Line 181-213, 254-259, 265-270, 276-284) to follow a logical progression discussing the fundamental technologies in proteomics relevant to the IPF research. Thus, we have emphasized technical features of proteomic approaches that have demonstrated translational potential in IPF, such as LC-MS, GC-MS quantitative proteomic etc., whose details and biomarkers identified will be discussed in next sections. Thus, techniques with limited or no current clinical applicability in IPF are now mentioned more briefly or deleted, respectively.
  • We also introduced a summarizing section 8 (Page 17, Line 672-703) that highlights the most promising protein biomarkers identified with Mass Spectrometry techniques for clinical translation research in IPF.

Some biomarkers including MMP-7, CCL18, periostin, KL-6, and surfactant proteins SP-A and SP-D are presented in several sections with almost identical wording. Rather than summarizing information, the authors duplicate it in plasma, BALF, tissue, and cellular subsections. Comparative synthesis is absent: the review could benefit from synthesized tables or figures stratifying biomarkers by validity status, reproducibility, and clinical usefulness, and not listing the same molecules again and again.

We thank the Reviewer for the comment. We wanted to discuss MMP-7, CCL18 KL-6 and surfactant proteins SP-A and SP-D in different sections since they have been involved in different proteomic studies from different biological sources and samples. Thus, we made up new figures that show the stratified biomarkers in different biological sources by their validity status, clinical relevance and the mass spectrometry technique used for their identification. In particular, we showed Biomarkers for Serum Plasma (Figure 2, Pag 10) BALF (Figure 3, Pag. 12) lung tissue (Figure 4, Pag.13) and cell line and primary cell (Figure 5 pag. 16). These visualizations aim to improve readability and help the reader easily understand which biomarkers are associated with each sample type.

Most importantly, the manuscript fails in critical appraisal. Most biomarker studies are reported uncritically, without mentioning sample size, study design, replication, or confounding variables. For instance, while KL-6 and MMP-7 are repeatedly highlighted as the most promising, no explanation is given for why they remain beyond standard clinical care after decades of research. Similarly, periostin and CCL18 are reported to be biomarkers predictive of IPF without illustrating their non-specificity or the difficulty of distinguishing IPF from other interstitial lung diseases.

We thank the reviewer for this critical and constructive comment. We fully agree that the original version of the manuscript lacked an adequate level of critical appraisal. In response, we have thoroughly revised the relevant sections to provide a more rigorous and evidence-based evaluation of discussed biomarker, explicitly addressing study design, sample size, replication, and confounding factors (Pag 10 Line 416-423, Pag 11 470-474, Pag 13 Line 519-522, Pag 15 Line 624-629). To address these concerns, we have also introduced a dedicated section number 8 (Pag 17, Line 671-700) and a summary Table 1 (Pag 17-18) showing the promising biomarkers identified form different biological sources such as KL-6 and MMP-7 that have not yet transitioned into standard clinical practice—emphasizing their strength and limitations.

The clinical translation sections are tenuous. The authors predict proteomics will improve diagnosis, prognosis, and therapy but fail to demonstrate how some biomarkers could realistically be used in everyday clinical practice. Scenarios such as differential diagnosis relative to other ILDs, prediction of response to antifibrotic therapy, or risk stratification for acute exacerbation are not addressed in depth. Without such context, the argument remains in the abstract and fails to convince the reader of clinical application.

We thank the Reviewer and we integrated the section number 10 “Clinical Translation of proteomics from bench to bedside in IPF” with the information’s highlighted by the Reviewer. In particular we discussed on how some biomarkers could realistically be used in everyday clinical practice facing different clinical needs such as differential diagnosis relative to other ILDs, prediction of response to antifibrotic therapy, or risk stratification for acute exacerbation (Pag 19 Line 765-780).

The inclusion of artificial intelligence in Section 8 is a weakness. The description of the account is shallow and complacent, portraying AI as a transformative device without calling out the largest dangers of reproducibility, interpretability, heterogeneity of data, and validation. There has to be some balance of discussion here; otherwise, the section reads more like an advertisement than a critical scholarship.

We thank the reviewer for this insightful observation. In response, we have revised Section 9 (Page 18-19, Line 744-754) to explicitly address key limitations of AI in biomarker research and clinical practice, including challenges related to reproducibility, interpretability, data heterogeneity, and the urgent need for external validation. This ensures a more balanced treatment of the topic.

Tables and figures can be done better: Figure 1 is essentially a generic textbook figure of proteomic workflows and not IPF-specific, and Table 1 is exhaustive but intimidating. Neither conveys to the reader at a glance which biomarkers have the greatest potential for translation. A reorganized table by strength of evidence, validation status, and reproducibility would be much more useful.

We thank the reviewer for the comment and we have modified the Figure 1 that shows the key technical aspects of top-down and bottom-up proteomic (Figure 1A-B Pag. 7), while we added other Figures to stratified the biomarkers in IPF from different biological source, as mentioned before. We changed Table 1 in Table 2 that describe all the biomarkers identified and cited throughout the Manuscript Review, while Table 1 shows the most relevant biomarkers identified across different proteomic studies and biological samples, with the greatest potential for translation.

There are a few issues of language and style as well. Typos, awkward phrasing, and overly verbose jargon sentences confuse a great deal. The manuscript could use a good close edit by a native English speaker, too.

We thank the Reviewer for the detailed and constructive feedback regarding the language. We have carefully revised the manuscript to correct all identified typographical errors and phrasing that the Reviewer mentioned in the new version.

Modena 14/10/2025

Last Author

Round 2

Reviewer 2 Report

Comments and Suggestions for Authors

The manuscript's clarity, organization, and clinical applicability have all greatly improved. A lot more effort has been put into the updated narrative, and the content is easier to follow thanks to the addition of figures and tables.

I propose adding more specific references or succinct numerical information for important studies, emphasizing the difference between exploratory and validated biomarkers, and more directly connecting the clinical translation section to diagnostic and therapeutic pathways in order to strengthen the paper even more.

The general readability would also be improved with a mild linguistic polish.

Comments on the Quality of English Language

I recommend a careful language polish by a native English speaker.

Author Response

On behalf of all the Authors we would like to thank all the Reviewers for their valuable time and useful contribution to our Manuscript Review “Predicting and treating pulmonary fibrosis with proteomic biomarker investigations”. We strongly appreciate the suggestions and inputs received that will definitely improve our manuscript Review. Please find below the detailed responses to Reviewer 2.

Reviewer 2 (minor revision)

The manuscript's clarity, organization, and clinical applicability have all greatly improved. A lot more effort has been put into the updated narrative, and the content is easier to follow thanks to the addition of figures and tables.

We thank the Reviewer for the feedback on our revised manuscript. Thanks to the comments of the Reviewers the Manuscript have improved the clarity and the overall organization.

I propose adding more specific references or succinct numerical information for important studies, emphasizing the difference between exploratory and validated biomarkers, and more directly connecting the clinical translation section to diagnostic and therapeutic pathways in order to strengthen the paper even more.

We thank the Reviewer for this insightful and constructive suggestion. Following the recommendation, we have added more detailed quantitative information regarding the number of patients and cohort sizes in the most relevant studies that identified IPF validated and exploratory biomarkers. Moreover, we have included additional references to strengthen the section and provide a broader overview of key studies in the field.
These revisions can be found on page 16, lines 642–675.

In addition, as suggested, we have explicitly linked the “Clinical Translation” section to diagnostic and therapeutic pathways, emphasizing how validated proteomic biomarkers can inform patient stratification, therapy selection, and disease monitoring.
These revisions can be found on page 17, lines 690–697.

The general readability would also be improved with a mild linguistic polish.

We sincerely thank the Reviewer for this valuable observation. Following the suggestion, the entire manuscript has undergone a comprehensive linguistic and stylistic revision to improve overall readability, clarity, and flow. Several sentences have been reformulated for better clarity and conciseness, while maintaining the original scientific meaning. All linguistic and stylistic changes are visible in the revised manuscript using track changes.

Comments on the Quality of English Language

I recommend a careful language polish by a native English speaker.

We sincerely thank the Reviewer for this valuable observation. Following the suggestion, the entire manuscript has undergone a comprehensive linguistic and stylistic revision to improve overall readability, clarity, and flow. Several sentences have been reformulated for better clarity and conciseness, while maintaining the original scientific meaning. All linguistic and stylistic changes are visible in the revised manuscript using track changes.

Modena 23/10/2025

Last Author